# AudioLCM: Efficient and High-Quality Text-to-Audio Generation with Minimal Inference Steps

### Huadai Liu
Zhejiang University
Shanghai Artificial Intelligence Laboratory
liuhuadai@zju.edu.cn

### Rongjie Huang
Zhejiang University
rongjiehuang@zju.edu.cn

### Yang Liu
Zhejiang University
22160155@zju.edu.cn

### Hengyuan Cao
Zhejiang University
caohy@zju.edu.cn

### Jialei Wang
Zhejiang University
3220101016@zju.edu.cn

### Xize Cheng
Zhejiang University
xizecheng@zju.edu.cn

### Siqi Zheng
Alibaba Group
zsq174630@alibaba-inc.com

### Zhou Zhao[†]
Zhejiang University
Shanghai Artificial Intelligence Laboratory
zhaozhou@zju.edu.cn

## ABSTRACT

Recent advancements in Latent Diffusion Models (LDMs) have propelled them to the forefront of various generative tasks. However, their iterative sampling process poses a significant computational burden, resulting in slow generation speeds and limiting their application in text-to-audio generation deployment. In this work, we introduce AudioLCM, a novel consistency-based model tailored for efficient and high-quality text-to-audio generation. Unlike prior approaches that address noise removal through iterative processes, AudioLCM integrates Consistency Models (CMs) into the generation process, facilitating rapid inference through a mapping from any point at any time step to the trajectory's initial point. To overcome the convergence issue inherent in LDMs with reduced sample iterations, we propose the Guided Latent Consistency Distillation with a multi-step Ordinary Differential Equation (ODE) solver. This innovation shortens the time schedule from thousands to dozens of steps while maintaining sample quality, thereby achieving fast convergence and high-quality generation. Furthermore, to optimize the performance of transformer-based neural network architectures, we integrate the advanced techniques pioneered by LLaMA into the foundational framework of transformers. This architecture supports stable and efficient training, ensuring robust performance in text-to-audio synthesis. Experimental results on text-to-audio generation and text-to-music synthesis tasks demonstrate that AudioLCM needs only 2 iterations to synthesize high-fidelity audios, while it maintains sample quality competitive with state-of-the-art

models using hundreds of steps. AudioLCM enables a sampling speed of 333x faster than real-time on a single NVIDIA 4090Ti GPU, making generative models practically applicable to text-to-audio generation deployment. Our extensive preliminary analysis shows that each design in AudioLCM is effective. [1] [2]

## CCS CONCEPTS

• **Applied computing** → **Sound and music computing**; • **Computing methodologies** → **Natural language generation**.

## KEYWORDS

Text-to-Audio Generation, Consistency Model, Latent Diffusion Model

**ACM Reference Format:**

Huadai Liu, Rongjie Huang, Yang Liu, Hengyuan Cao, Jialei Wang, Xize Cheng, Siqi Zheng, and Zhou Zhao[†]. 2024. AudioLCM: Efficient and High-Quality Text-to-Audio Generation with Minimal Inference Steps. In *Proceedings of the 32nd ACM International Conference on Multimedia (MM '24), October 28-November 1, 2024, Melbourne, VIC, AustraliaProceedings of the 32nd ACM International Conference on Multimedia (MM'24), October 28-November 1, 2024, Melbourne, Australia.* ACM, New York, NY, USA, 10 pages. https://doi.org/10.1145/3664647.3681072

## 1 INTRODUCTION

Text-to-audio generation (TTA) [13, 20, 23] is a subfield of generative tasks that produces natural and precise audio from textual prompts. TTA has a wide range of applications, including sound effects creation, musical compositions, and synthesized speech. It is used in various domains, such as film post-production, video game development, and audio manipulation. Previous iterations of neural TTA models have been primarily categorized into two main types: language models [21, 29] and diffusion models [7, 12].

---

[1]Audio samples are available at https://AudioLCM.github.io/.
[2]Code is Available at https://github.com/Text-to-Audio/AudioLCM

(1) Language models encode raw waveform data into discrete representations and employ autoregressive language models to predict audio tokens based on textual input features. (2)Diffusion models: This category involves generating audio representations, such as discrete codes [50] or mel-spectrograms [13], from textual prompts using diffusion models. These representations are then converted into audio waveforms using separately trained vocoders. Although both methods have demonstrated the ability to produce high-quality audio samples, they are often limited by high computational costs, which can make it difficult to achieve both quality and efficiency in audio generation.

Diffusion models [9, 16, 41], particularly Latent Diffusion models (LDMs) [33, 34], have yielded unprecedented breakthroughs across various fields including image synthesis [32, 34, 35], video generation [11, 39], and audio synthesis [17, 24, 54]. A key feature of diffusion models is their iterative sampling mechanism, which progressively refines random initial vectors, thereby mitigating noise and enhancing sample quality iteratively. This iterative refinement process facilitates a flexible trade-off between computational resources and sample quality; typically, allocating additional compute for more iterations results in the generation of higher-quality samples. However, the generation procedure of diffusion models typically requires more computational costs for sample generation, causing slow inference and limited real-time applications.

To alleviate this computational bottleneck, advanced numerical solvers [27, 53] of Stochastic Differential Equations (SDE) or Ordinary Differential Equations (ODE) substantially reduce the required Number of Function Evaluations (NFE), further improvements are challenging due to the intrinsic discretization error present in all differential equation solvers. Recent endeavors have focused on enhancing sample efficiency through distillation models. These models [30, 31, 36] aim to distill the knowledge from a pre-trained diffusion model into compact architectures capable of conducting inference with minimal computational overhead. For instance, Meng et al. [31] proposed a two-stage distillation approach aimed at improving the sample efficiency of classifier-free guided models. However, this approach presents several challenges: (1) Computational Intensity: As estimated by Liu et al. [26], the distillation process entails significant computational resources, necessitating at least 45 A100 GPU days for training 2-step student models. (2) Error Accumulation: The two-stage guided distillation process may inadvertently introduce accumulated errors, thereby compromising the performance of the distilled models, resulting in suboptimal results.

In this work, we propose AudioLCM, a consistency model based method tailored for efficient and high-quality text-to-audio generation; 1) To avoid significant degradation of perceptual quality when reducing reverse iterations, AudioLCM integrates Consistency Models (CMs) [40] into the generation process, facilitating rapid inference through a mapping from any point at any time step to the trajectory's initial point; 2) To overcome the convergence issue inherent in LDMs with reduced sample iterations and accelerate the convergence speed, AudioLCM reduces the data variance in the target side via One-stage Guided Consistency Distillation with multi-step Ordinary Differential Equation (ODE) solver. Specifically, We can sample $x_{t+k}$ from the transition density of the SDE, and

then compute $\hat{x}^{\phi}$ using $k$ ($k > 1$) discretization step of the numerical ODE solver. We also include the Classifier-free Guidance (CFG) [10] into the process of distillation and sampling to improve the audio quality.

Moreover, as a monochromatic image, the mel-spectrogram lacks spatial translation invariance. The vertical axis of the mel-spectrogram corresponds to the frequency domain, implying that mel-spectrogram patches at different vertical positions can convey completely different information and should not be treated equivalently. In addition, the use of a 2D convolution layer and a spatial transformer-stacked U-Net architecture limits the model's ability to generate audio of variable length. Prior research [12] in TTA has demonstrated the effectiveness of addressing this issue. To enhance the performance of transformer-based neural network architectures, we integrate advanced techniques developed by LLaMA [45] into the foundational framework of transformers. The LLaMA methodology, characterized by its causal transformer architecture tailored for large language models, holds promise for enhancing the capabilities of transformer backbones. By seamlessly integrating these methodologies, AudioLCM aims to unlock heightened performance levels and facilitate a more stable training process.

Experimental results from text-to-audio generation and text-to-music synthesis tasks demonstrate that AudioLCM achieves high-fidelity audio synthesis in only 2 iterations, while maintaining sample quality comparable to state-of-the-art models that require hundreds of steps. In addition, AudioLCM achieves a sampling rate 333 times faster than real-time on a single NVIDIA 4090Ti GPU, making generative models practical for text-to-speech applications. Our comprehensive preliminary analysis demonstrates the effectiveness of each component within AudioLCM. The main contributions of this study are summarized below:

- We propose AudioLCM, a novel consistency-based model for efficient and high-quality text-to-audio generation. Unlike prior approaches that address noise removal through iterative processes, AudioLCM maps from any point at any time step to the trajectory's initial point. To overcome the convergence issue inherent in LDMs with reduced sample iterations, AudioLCM proposes the One-stage Guided Latent Consistency Distillation with a multi-step ODE solver, shortening the time schedule from thousands to dozens of steps while maintaining sample quality.

- AudioLCM enhances transformer-based neural network architectures by integrating advanced techniques pioneered by LLaMA. These techniques, renowned for their tailored causal transformer architecture, promise to deliver improved performance and training stability.

- Experimental results demonstrate that AudioLCM can synthesize high-fidelity audio samples with only 2 iterations, while maintaining sample quality that is competitive with state-of-the-art models that employ hundreds of steps. This makes generative models practically applicable for text-to-audio generation deployment.

## 2 PRELIMINARIES

In this section, we briefly introduce the theory of Diffusion models and consistency models.

## 2.1 Diffusion Models

Diffusion models [14, 42], also known as score-based generative models, are designed to generate data by iteratively perturbing data with Gaussian noise and subsequently generating samples from the perturbed data using a reverse denoising process. Let $p_{data}(x)$ denote the original data distribution. Continuous-time diffusion models [15, 18, 43] conduct the forward process using a stochastic differential equation (SDE):

$$dx_t = f(x_t, t)dt + g(t)dw_t, t \in [0, T]. \qquad (1)$$

where $f()$ and $g$ represent the drift and diffusion coefficients, respectively, while $w_t$ denotes the standard Brownian motion.

By considering the existence of an ordinary differential equation (ODE), also called the *Proability Flow ODE (PF-ODE)*:

$$\frac{dx_t}{dt} = f(x_t, t) - \frac{1}{2}g(t)^2 \nabla_x log p_t(x_t). \qquad (2)$$

where $\nabla log p_t(x)$ is the score function of $p_t(x)$.

We train the noise prediction model $\epsilon_\theta(x_t, t)$ to approximate the score function, enabling us to derive the following empirical PF-ODE:

$$\frac{dx_t}{dt} = f(x_t, t) + \frac{g(t)^2}{2\sigma_t}\epsilon_\theta(x_t, t), x_T \in \mathcal{N}(0, \sigma^2 I) \qquad (3)$$

## 2.2 Consistency Models

Consistency models [40] (CMs) represent a novel class of models facilitating few-step or even one-step generation. The fundamental principle underlying consistency models is the establishment of a mapping capable of connecting any point at any given timestep to the starting point of a trajectory. This mapping is encapsulated by the *consistency function*, denoted as $f : (x_t, t) \rightarrow x_\epsilon$, where $x_t$ represents a predetermined trajectory of the PF-ODE and $\epsilon$ is a predefined small positive value. A distinctive feature of the consistency function is its *self-consistency* property, where points on the same trajectory converge to the same starting point. This property is expressed mathematically as follows:

$$f(x_t, t) = f(x_{t'}, t'), \forall t, t, \in [\epsilon, T]. \qquad (4)$$

The objective of a consistency model $f_\theta$ is to estimate the consistency function $f$ from data, thereby enforcing the self-consistency property. For any consistency function, it holds that $f(x_\epsilon, \epsilon) = x_\epsilon$, termed as the *boundary condition*. To satisfy this boundary condition, the consistency model $f_\theta$ is parameterized as follows:

$$f_\theta = c_{skip}(t)x + c_{out}(t)F_\theta(x, t), \qquad (5)$$

where $c_{skip}(t)$ and $c_{out}(t)$ are differentiable functions, with $c_{skip}(\epsilon) = 1$ and $c_{out}(\epsilon) = 0$. Additionally, $F_\theta(\cdot, \cdot)$ represents a deep neural network. Consistency models can be trained either by *consistency distillation* or from scratch.

In the former scenario, consistency models distill insights from pre-trained diffusion models into a target neural network tailored for sampling. Given a data point $x$, an accurate estimation of $x_{t_n}$ from $x_{t_{n+1}}$ is achieved by executing a single discretization step using a numerical ODE solver. Mathematically, this is expressed as:

$$\hat{x}_{t_n}^\phi := x_{t_{n+1}} + (t_n - t_{n+1})\Phi(x_{t_{n+1}}, t_{n+1}; \phi), \qquad (6)$$

where $\Phi(\cdot, \cdot; \phi)$ signifies a one-step ODE solver applied to the empirical PF-ODE. Afterward, the objective is to minimize discrepancies within the pair $(\hat{x}_{t_n}^\phi, x_{t_{n+1}})$ by *consistencydistillationloss* defined as:

$$\mathcal{L}(\theta, \theta^-; \Phi) = \mathbb{E}_{x,t}[d(f_\theta(x_{t_{n+1}}, t_{n+1}), f_{\theta^-}(\hat{x}_{t_n}^\phi, t_n))]. \qquad (7)$$

where $\hat{x}_{t_n}^\phi$ corresponds to Equation 6, $\theta^-$ represents a running average of past $\theta$ values during the optimization process, and $d(\cdot, \cdot)$ denotes a metric function used to quantify the distance between the pair.

## 3 AUDIOLCM

This section presents our proposed *AudioLCM*, a few-step and one-step consistency model for high-fidelity text-to-audio generation. The framework of the proposed method is shown in Figure 1. We begin by outlining the motivation behind each design choice in AudioLCM. Next, we detail the process of selecting a diffusion teacher model and performing Guided Consistency Distillation from it. We then discuss the model architecture and training loss used in AudioLCM, followed by a discussion of the training and inference algorithms.

## 3.1 Motivation

Diffusion models have made significant progress in various domains such as audio and image generation. However, several challenges hinder their industrial deployment: 1) The iterative nature of the prevailing latent diffusion text-to-audio models requires a considerable amount of computational resources for sample generation, resulting in slow inference times and limited real-time applicability. This limitation is a barrier to the practical use of these models in real-world scenarios. 2) Diffusion models typically train noise prediction models using extended time-step schedules. For example, models such as AudioLDM employ a time-step schedule of length 1,000, resulting in slower convergence rates and increased computational requirements.

Recently, Consistency Models [40] have emerged as a novel generative approach to support few-step and even one-step generation. Unlike previous methods that rely on iterative processes for noise removal, AudioLCM integrates consistency models (CMs) into the generation process. This integration enables rapid inference by mapping any point at any time step to the starting point of the trajectory. To address the convergence challenges inherent in Latent Diffusion Models [34] (LDMs) with reduced sample iterations, we introduce Guided Consistency Distillation with a multi-step ODE solver. This approach significantly reduces the schedule from thousands to tens of steps while maintaining sample quality, resulting in faster convergence and high-quality generation. Furthermore, to improve the performance of transformer-based neural network architectures, we integrate advanced techniques pioneered by LLaMA into the basic framework of transformers.

In conclusion, AudioLCM leverages a consistency model in the audio latent space to achieve rapid convergence and generation while ensuring sample quality. This approach is suitable for interactive, real-world applications.

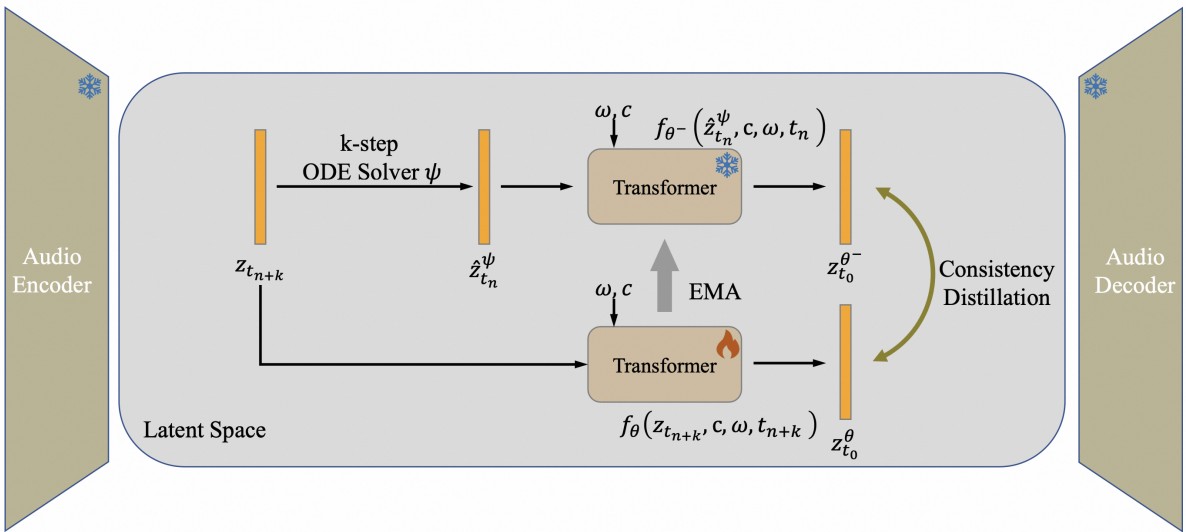

**Figure 1: An illustration of AudioLCM. AudioLCM propose the Guided Consistency Distillation with $k$-step ODE solver. $c$ is the text embedding and $\omega$ is the classifier-free guidance scale.**

## 3.2 Teacher model

In this subsection, we describe the teacher model we select. As a blossoming class of generative models, many text-to-audio generation systems apply latent diffusion models to generate high-quality audio. However, the teacher model is supposed to achieve fast and high-quality text-to-audio generation, and thus the distilled student could inherit its powerful capability. As such, we choose the Make-An-Audio 2 [12] and modify its feed-forward transformer [47] with our enhanced transformer-based Backbone while keeping other designs as our teacher model to strike a proper balance between perceptual quality and sampling speed.

***Enhanced Transformer-based Backbone***. Previous research [12] on TTA synthesis treated the mel-spectrogram as a single-channel image, similar to text-to-image synthesis, but the mel-spectrogram lacks spatial translation invariance due to its representation of the frequency domain. Consequently, patches at different heights carry distinct meanings and should not be treated equally. Furthermore, the use of a 2D-convolution layer and spatial transformer-stacked U-Net architecture limits the model's ability to generate variable-length audio. According to the practice of Make-An-Audio 2, they use a modified audio VAE that uses a 1D-convolution-based model and a feed-forward Transformer-based diffusion denoiser backbone. Inspired by the success achieved by LLaMA [45], we integrate the advanced techniques pioneered by LLaMA into the foundational framework of feed-forward transformers. We draw the main improvement with the original architecture:

- **Pre-normalization**: We use the RMSNorm normalizing function [52] and normalize the input of each transformer layer for the training stability.
- **Rotary Embeddings**: We replace the absolute positional embeddings with rotary positional embeddings [44] (RoPE) at each layer of transformer.

- **SwiGLU Activation**: We remove the SiLU function, and instead add the SwiGLU activation function [38] to improve the performance.

## 3.3 Guided Consistency Distillation

Given the audio latent space established by the teacher model, aimed at mitigating computational overhead and enhancing performance, we proceed to redefine the PF-ODE governing the reverse diffusion process, denoted as Equation 3:

$$\frac{dz_t}{dt} = f(z_t, t) + \frac{g(t)^2}{2\sigma_t}\epsilon_\theta(z_t, c, t), z_T \in \mathcal{N}(0, \sigma^2 I) \qquad (8)$$

To preserve the property of self-consistency, we introduce the consistency function $f_\theta : (z_t, c, t) \to z_0$, which maps points from any trajectory of the PF-ODE to the origin of the trajectory. We then parameterize the consistency function using the noise prediction model $\hat{\epsilon}\theta$ to satisfy the boundary condition:

$$f_\theta(z, c, t) = c_{skip}(t)z + c_{out}(t)(\frac{z - \sigma_t\hat{\epsilon}_\theta(z, c, t)}{\alpha_t}). \qquad (9)$$

where $c_{skip}(0) = 1$, $c_{out}(0) = 0$, $\alpha_t$ and $\sigma_t$ specify the noise schedule, and $\hat{\epsilon}_\theta(z, c, t)$ is a noise prediction model initialized with the parameters of the teacher model.

To adapt to the discrete-time schedule in the teacher model, we utilize DDIM as the ODE solver $\Psi(z_t, t, c)$ to obtain an accurate estimate of $z_{t_n}$ from $z_{t_{n+1}}$. Note that we only use the solver in training, not in inference. Then we estimate the evolution of the PF-ODE from $t_{n+1} \to t_n$ using one-step ODE solver:

$$\hat{z}_{t_n}^\Psi - z_{t_{n+1}} = \int_{t_{n+1}}^{t_n} (f(z_t, t) + \frac{g^2(t)}{2\sigma_t}\epsilon_\theta(z_t, c, t))dt. \qquad (10)$$

Finally, we calculate the consistency distillation loss to optimize the AudioLCM model:

$$\mathcal{L}_{CD}(\theta, \theta^-; \Psi) = \mathbb{E}_{z,c,t}[d(f_\theta(z_{t_{n+1}}, c, t_{n+1}), f_{\theta^-}(\hat{z}_{t_n}^\Psi, c, t_n))]. \qquad (11)$$

***Classifier-free Guidance (CFG).*** The effectiveness of classifier-free guidance has been demonstrated in the synthesis of high-quality text-aligned audio [10]. To facilitate one-stage guided distillation, we integrate CFG into the distillation process using an extended PF-ODE. Specifically, we replace the noise prediction model $\hat{\epsilon}\theta(z, c, t)$ with $\hat{\epsilon}\theta(z, \omega, c, t)$, which incorporates the learnable guidance scale embedding $\omega$. Consequently, we introduce an extended consistency function $f_\theta : (z_t, w, c, t) \rightarrow z_0$:

$$f_\theta(z, w, c, t) = c_{skip}(t)z + c_{out}(t)(\frac{z - \sigma_t\hat{\epsilon}_\theta(z, w, c, t)}{\alpha_t}). \quad (12)$$

Similarly, the consistency loss remains the same as Equation 11, with the only difference of consistency function.

## 3.4 Accelerating Distillation with Multi-step ODE solver

Latent diffusion models typically train noise prediction models with a long time-step schedule to achieve high-quality generation results. The formula of the teacher model has the same problem, which needs to sample across all 1,000 timesteps. However, the extended time steps are highly time-consuming and computer-consuming for applying the guided consistency distillation. Consistency models conduct one-step ODE solvers to estimate $z_{t_n}$ from $z_{t_{n+1}}$, while they are close to each other and incurring a small consistency loss. To achieve fast convergence while preserving generation quality, we introduce multi-step ODE solvers to considerably shorten the length of time schedule (from thousands to dozens).

Instead of ensuring consistency between adjacent time steps $t_{n+1} \rightarrow t_n$, we perform a k-step discretization step of the ODE solver to obtain $z_{t_n}$ from $z_{t_{n+k}}$. The small $k$ step leads to slow convergence while very large $k$ may incur large approximation errors of the ODE solvers. Therefore, it is critical to choose a proper $k$ which achieves fast convergence while preserving the sample quality. Results in Section 4.2 show the effect of different $k$ values and reveal that the multi-step ODE solvers are crucial in accelerating the distillation process. Specifically, $\hat{z}_{t_n}^\Psi$ is an estimate of $z_{t_n}$ using numerical multi-step ODE solvers:

$$\hat{z}_{t_n}^\Psi - z_{t_{n+k}} = (1+\omega)\Psi(z_{t_{n+k}}, t_{n+k}, t_n, c) - \omega\Psi(z_{t_{n+k}}, t_{n+k}, t_n, \emptyset) \quad (13)$$

The modification of the consistency distillation loss in Equation 9 ensures consistency from $t_{n+k}$ to $t_n$.

$$\mathcal{L}_{CD}(\theta, \theta^-; \Psi) = \mathbb{E}_{z,c,t}[d(f_\theta(z_{t_{n+k}}, c, \omega, t_{n+k}), f_{\theta^-}(\hat{z_{t_n}}_{t_n}^\Psi, c, \omega, t_n))]. \quad (14)$$

## 3.5 Training and Inference Procedures

The training and sampling algorithms of AudioLCM have been outlined in Algorithm 1 and Algorithm 2, respectively.

***Training.*** All consistency models are initialized with their corresponding pre-trained diffusion models. To ensure a more robust model fit, we employ the Huber loss $\mathcal{L}_\eta$ as the distance function in Equation 15:

$$L_\eta(y, f(x)) = \begin{cases} \frac{1}{2}(y - f(x))^2, & \text{if } |y - f(x)| \le \eta \\ \eta(|y - f(x)| - \frac{1}{2}\eta), & \text{otherwise} \end{cases} \quad (15)$$

where $\eta$ is a hyperparameter used to regulate the sensitivity of the loss function to outliers.

---

**Algorithm 1** Guided Consistency Distillation with Multi-Step ODE Solver

1: **Input**: dataset $\mathcal{D}$, initial model parameter $\theta$ from teacher model, learning rate $\varphi$, ODE solver $\Psi(\cdot, \cdot, \cdot, \cdot)$, distance function $d(\cdot, \cdot)$, EMA rate $\mu$, noise schedule $\alpha(t), \sigma(t)$, guidance scale $[\omega_{min}, \omega_{max}]$, multi-step value $k$, and encoder $E(\cdot)$ Encoding training data into latent space: $\mathcal{D}_z = (z, c)|z = E(x), (x, c) \in \mathcal{D}$
2: $\theta^- \leftarrow \theta$
3: **repeat**
4:     Sample $(z, c) \sim \mathcal{D}_z$, $n \sim \mathcal{U}(1, N - k)$ and $w \sim [\omega_{min}, \omega_{max}]$
5:     Sample $z_{t_{n+k}} \sim \mathcal{N}(\alpha(t_{n+k})z; \sigma^2(t_{n+k})\mathbf{I})$
6:     $\hat{z}_{t_n}^\Psi \leftarrow z_{t_{n+k}} + (1 + \omega)\Psi(z_{t_{n+k}}, t_{n+k}, t_n, c) - \omega\Psi(z_{t_{n+k}}, t_{n+k}, t_n, \emptyset)$
7:     $\mathcal{L}(\theta, \theta^-; \Psi) \leftarrow d(f_\theta(z_{t_{n+k}}, c, \omega, t_{n+k}), f_{\theta^-}(\hat{z_{t_n}}_{t_n}^\Psi, c, \omega, t_n))$
8:     $\theta \leftarrow \theta - \varphi\nabla\mathcal{L}(\theta, \theta^-)$
9:     $\theta^- \leftarrow \text{stopgrad}(\mu\theta^- + (1 - \mu)\theta)$
10: **until** AudioLCM converged

---

***Inference.*** During inference, AudioLCM generates samples by first sampling from the initial distribution $\hat{z}T \sim \mathcal{N}(0, T^2\mathbf{I})$ and then evaluating the consistency model for $z = f\theta(\hat{z}_T, \omega, c, T)$. The LCM models can be evaluated with either a single forward pass or multiple passes, alternating denoising and noise injection steps to enhance sample quality. As outlined in Algorithm 2, this multi-step sampling approach offers the flexibility to balance computational resources with sample quality. Ultimately, the generated audio latents $z$ are transformed into mel-spectrograms and subsequently used to generate waveforms using a pre-trained vocoder.

---

**Algorithm 2** Multi-step Consistency Sampling

1: **Input**: AudioLCM $f_\theta$, Sequence of timesteps $\tau_1 > \tau_2 > \cdots > \tau_{N-1}$, text condition $c$, Guidance Scale $\omega$, Noise Schedule $\alpha(t), \sigma(t)$, VAE Decoder $D(\cdot)$, and Vocoder $V(\cdot)$.
2: Sample $z_T \sim \mathcal{N}(0, \mathbf{I})$
3: $z \leftarrow f_\theta(\hat{z}_T, \omega, c, T)$
4: **for** $n = 1$ to $N - 1$ **do**
5:     $\hat{z}_{\tau_n} = \mathcal{N}(\alpha(\tau_n)z; \sigma^2(\tau_n)\mathbf{I})$
6:     $z \leftarrow f_\theta(\hat{z}_{\tau_n}, \omega, c, \tau_n)$
7: **end for**
8: **return** $V(D(z))$

---

# 4 EXPERIMENTS

## 4.1 Experimental setup

***Dataset.*** For text-to-sound generation, we use a diverse combination of datasets to train our teacher model, while the Audio-Caps dataset [19] is specifically utilized for training our AudioLCM model. For text-to-music generation, we exclusively employ the LP-Musicaps dataset for both teacher and AudioLCM training endeavors. More details about dataset can be found in Appendix A.

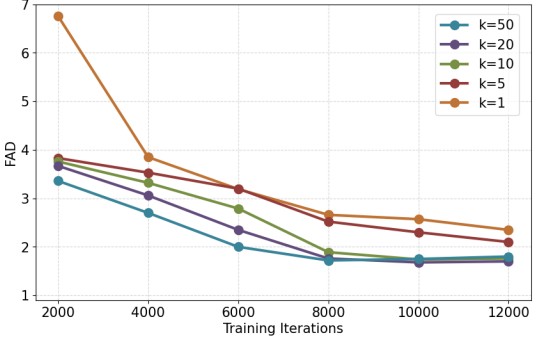

(a) Preliminary Analyses on Multi-Step ODE Solver.

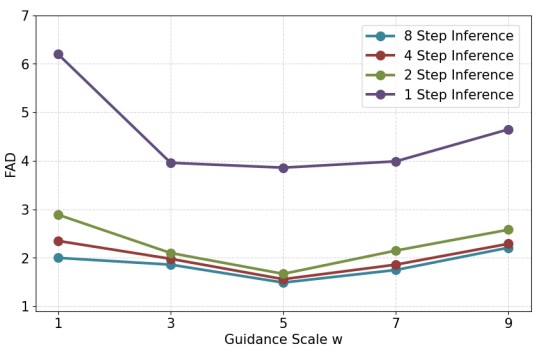

(b) Preliminary Analyses on Classifier-free Guidance.

**Figure 2: In subfigure (a), we assess the correlation between the audio quality and the estimate interval $k$ of ODE solver across the test set. In subfigure (b), we delves into the examination of how different scales of classifier-free guidance contribute to the overall performance of FAD.**

***Model configurations.*** Our teacher model is originally trained with $\epsilon$-prediction, we utilize its pre-trained VAE, a continuous 1D-convolution-based network. This VAE is used to compress the mel-spectrogram into a 20-channel latent representation with a temporal axis downsampling rate of 2. Training AudioLCM involves 15,000 iterations on an NVIDIA 4090Ti GPU, with a batch size of 8 per GPU. We use the AdamW optimizer with a learning rate of 9.6e-5 and an exponential moving average (EMA) rate of $\mu = 0.95$. For the ODE solver, we use the DDIM solver with a multi-step parameter $k = 20$. The control scale range is defined as $[\omega_{min}, \omega_{max}] = [4, 12]$. Our choice for the vocoder is BigVGAN [22], which is known for its universal applicability to different scenarios. We train the vocoder on the AudioSet dataset to ensure robust performance. More details on the model configuration can be found in Appendix B.

***Evaluation Metrics.*** Our models conduct a comprehensive evaluation [5] using both objective and subjective metrics to measure audio quality, text-audio alignment fidelity, and inference speed. Objective assessment includes Kullback-Leibler (KL) divergence, Frechet audio distance (FAD), and CLAP score to quantify audio quality. The Real-time Factor (RTF) is also introduced to measure the system's efficiency in generating audio for real-time applications. RTF is the ratio between the total time taken by the audio system to synthesize an audio sample and the duration of the audio. In terms of subjective evaluation, we conduct crowd-sourced human assessments employing the Mean Opinion Score (MOS) to evaluate both audio quality (MOS-Q) and text-audio alignment faithfulness (MOS-F). Detailed information regarding the evaluation procedure can be accessed in Appendix C.2.

## 4.2 Preliminary Analyses

We conduct a comparative analysis of the ODE Solver with varying estimate step $k$ and explore the impact of the guidance scale $\omega$ in LCM. To assess the Multi-step ODE schedule, we evaluate the convergence speed and identify the optimal performance among different $k$ values using training iterations and Frechet audio distance (FAD). In analyzing classifier-free guidance, we examine the sample quality across different inference steps using the guidance scale and FAD.

***Multi-step ODE solver.*** The results are shown in Figure 2(a). Several observations emerge from these results: 1) Compared to the one-step ODE solver (k=1), the multi-step ODE solver exhibits significantly faster convergence, underscoring the effectiveness of using multiple steps in accelerating convergence speed. 2) Increasing the k value from 1 to 20 results in faster convergence and improved performance. However, using a larger step size, such as k = 50, may lead to suboptimal results. Therefore, we choose k = 20 to strike a balance between sample quality and convergence speed.

***Classifier-free Guidance.*** The results are summarized and presented in Figure 2(b). Here are our main observations: 1) As expected, larger inference steps show superior performance compared to smaller steps of the same size, highlighting the effectiveness of multi-step sampling in improving sample quality. 2) The performance of the Classifier-Free Guidance (CFG) model peaks at a certain value. Specifically, increasing the scale from 1 to 5 significantly enhances audio quality. However, increasing the scale further to 9 results in a gradual decline in performance. Therefore, we opt for a scale of 5 as a guideline. 3) The performance disparities among inference steps of 2, 4, and 8 are minimal, indicating the effectiveness of LCM in the 2-8 step range. However, a significant performance gap is evident in one-step inference, suggesting areas for potential improvement.

## 4.3 Performance on Text-to-Audio Generation

We conduct a comparative analysis of the quality of generated audio samples and inference latency across various systems, including GT (i.e., ground-truth audio), AudioGen, Make-An-Audio, AudioLDM-L, TANGO, Make-An-Audio 2, ConsistencyTTA, and AudioLDM 2, utilizing the published models as per the respective paper and the same inference steps of 100 for a fair comparison. The evaluations are conducted using the AudioCaps test set and then calculate the objective and subjective metrics. The results are compiled and presented in Table 1. From these findings, we draw the following conclusion:

**Audio Quality** In terms of audio quality, our proposed system, AudioLCM, demonstrates outstanding performance, particularly when configured with 4 inference steps. With a Fréchet Audio Distance (FAD) of 1.56 and Kullback-Leibler Divergence (KL) of 1.30,

| Model | NFE | Objective Metrics | | | | Subjective Metrics | |
|-------|-----|---------|---------|----------|----------|---------|---------|
| | | FAD (↓) | KL (↓) | CLAP (↑) | RTF (↓) | MOS-Q(↑) | MOS-F(↑) |
| GT | / | / | / | 0.670 | / | 86.65 | 84.23 |
| AudioGen-Large | / | 1.74 | 1.43 | 0.601 | 1.890 | / | / |
| Make-An-Audio | 100 | 2.45 | 1.59 | 0.616 | 0.280 | 70.32 | 66.24 |
| AudioLDM | 200 | 4.40 | 2.01 | 0.610 | 1.543 | 64.21 | 60.96 |
| Tango | 200 | 1.87 | 1.37 | 0.650 | 1.821 | 74.35 | 72.86 |
| AudioLDM 2 | 100 | 1.90 | 1.48 | 0.622 | 1.250 | / | / |
| AudioLDM 2-Large | 100 | 1.75 | 1.33 | 0.652 | 2.070 | 75.86 | 73.75 |
| Make-An-Audio 2 | 100 | 1.80 | 1.32 | 0.645 | 0.170 | 75.31 | 73.44 |
| ConsistencyTTA | 2 | 2.65 | 18.76 | 0.618 | 0.004 | 73.26 | 70.22 |
| Teacher | 100 | **1.56** | **1.30** | **0.655** | 0.190 | **78.67** | **76.19** |
| **AudioLCM** | 2 | 1.67 | 1.37 | 0.617 | **0.003** | 77.39 | 75.02 |

**Table 1: The audio quality and sampling speed comparisons. The evaluation is conducted on a server with 1 NVIDIA 4090Ti GPU and batch size 1. NFE (number of function evaluations) measures the computational cost, which refers to the total number of times the denoiser function is evaluated during the generation process.**

AudioLCM demonstrates minimal spectral and distributional discrepancies between the generated audio and ground truth. Even with only 2 inference steps, AudioLCM maintains competitive audio quality, showcasing its efficacy in generating high-fidelity audio samples. Although the CLAP score of 0.617 for AudioLCM is slightly lower than the state-of-the-art model's score of 0.652, it still indicates excellent text-audio alignments. Human evaluation results further confirm the superiority of AudioLCM, with MOS-Q and MOS-F scores of 77.39 and 75.02, respectively, surpassing AudioLDM 2-Large in subjective assessment. These findings suggest a preference among evaluators for the naturalness and faithfulness of audio synthesized by our model over baseline approaches. **Sampling Speed** AudioLCM boasts an efficient sampling process,

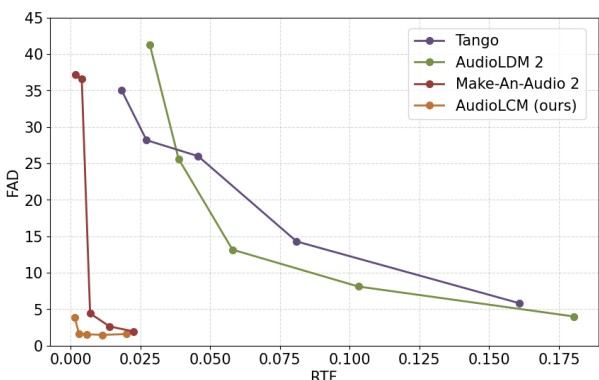

**Figure 3: We evaluate the relationship between the inference latency and sample quality measured by FAD.**

requiring only **2** iterations to synthesize high-fidelity audio samples, which translates to a remarkable speed of 333x times faster than real-time on a single NVIDIA 4090Ti GPU. This significant reduction in inference time positions AudioLCM as a leading solution, substantially outperforming dominant diffusion-based models.

In order to deepen our understanding of the correlation between inference latency and audio quality, we visualize the performance metrics, particularly focusing on FAD, concerning Make-An-Audio 2, TANGO, and AudioLDM 2. Our analysis, depicted in Figure 3,

reveals several key insights: 1) Diffusion-based models exhibit considerable performance degradation as inference time is shortened (i.e., the number of sampling steps is halved). In contrast, AudioLCM demonstrates minimal degradation until the RTF drops to 0.0015 (i.e., one-step inference), highlighting its robustness in achieving high-quality audio generation even with few inference steps. 2) Among the compared models, Make-An-Audio 2 demonstrates superior performance under low latency conditions, underscoring its suitability as a teacher model for knowledge distillation purposes. The distilled student models can inherit the powerful capabilities of Make-An-Audio 2, thereby enhancing overall model performance. 3) After approximately 10 steps, the trade-off between sample quality and inference speed inherent in consistency models becomes less apparent. This phenomenon arises due to the accumulation of discrete errors throughout the multi-step sampling process, leading to deviations of the generated audios from the target distribution.We defer the resolution of this challenge to future investigations and refinements.

**Zero-shot Evaluation** To further investigate the generalization performance of the models, we additionally test the performance of the models on the Clotho-evaluation dataset in the zero-shot setting. As illustrated in Table 2, AudioLCM with 4 sampling steps has significantly better results than AudioLDM 2-Large and Make-An-Audio 2 in IS and FAD, attributing to the scalability in terms of data usage. Furthermore, our 2 steps model continues to demonstrate competitive performance with baselines.

| Model | NFE | FAD↓ | KL↓ | IS↑ | RTF↓ |
|-------|-----|------|-----|-----|------|
| TANGO | 200 | 3.61 | 2.59 | 6.77 | 0.93 |
| AudioLDM 2-Large | 100 | 3.40 | 2.55 | 7.51 | 2.12 |
| Make-An-Audio 2 | 100 | 2.23 | 2.52 | 8.5 | 0.18 |
| Teacher | 100 | **2.19** | **2.49** | **10.14** | 0.20 |
| **AudioLCM** | 2 | 2.34 | 2.58 | 9.56 | **0.0006** |

**Table 2: Zero-shot generation results. We compare with Make-An-Audio 2, AudioLDM 2-Large, and Tango on Clotho-eval datasets. Inception score(IS) is used to evaluate both the quality and diversity of generated audio.**

| Model | NFE | Objective Metrics | | | | Subjective Metrics | |
| | | FAD (↓) | KL (↓) | CLAP (↑) | RTF (↓) | MOS-Q(↑) | MOS-F(↑) |
|---|---|---|---|---|---|---|---|
| GroundTruth | / | / | / | 0.46 | / | 88.42 | 90.34 |
| Riffusion | / | 13.31 | 2.10 | 0.19 | 0.40 | 76.11 | 77.35 |
| Mousai | / | 7.50 | / | / | / | / | / |
| Melody | / | 5.41 | / | / | / | / | / |
| MusicLM | / | 4.00 | / | / | / | / | / |
| MusicGen | / | 4.50 | 1.41 | 0.42 | 1.28 | 80.74 | 83.70 |
| MusicLDM | 200 | 5.20 | 1.47 | 0.40 | 1.40 | 80.51 | 82.35 |
| AudioLDM 2 | 200 | 3.81 | 1.22 | **0.43** | 2.20 | 82.24 | 84.35 |
| Teacher | 100 | **3.72** | **1.20** | 0.42 | **0.180** | **83.76** | **86.12** |
| **AudioLCM** | 2 | 3.92 | 1.24 | 0.40 | **0.003** | 82.29 | 84.44 |

Table 3: The comparison between AudioLCM and baseline models on the MusicCaps Evaluation set. We borrow the results of Mousai, Melody, MusicLM from the MusicGen [4].

## 4.4 Text-to-Music Generation

In this section, we conduct a comparative analysis of the audio samples generated by AudioLCM against a range of established Music Generation systems. These include: 1) GT, the ground-truth audio; 2) MusicGen [4]; 3) MusicLM [1]; 4) Mousai [37]; 5) Riffusion [6]; 6) MusicLDM [3]; 7) AudioLDM 2 [25]. The results are presented in Table 3, and we have the following observations: 1) In terms of audio quality, our teacher model consistently outperforms all diffusion-based methods and language models across a spectrum of both objective and subjective metrics. Furthermore, AudioLCM demonstrates competitive performance even when compared to diffusion models that require a significantly greater number of iterations, while also surpassing auto-regressive models. This underscores AudioLCM's effectiveness in producing high-quality music samples, establishing it as a highly capable model in the realm of audio synthesis. 2) In terms of sampling speed, AudioLCM stands out for its exceptional efficiency. It requires a mere 2 iterations to produce high-fidelity music samples, illustrating its potent capability to strike an optimal balance between the quality of the samples and the time required for inference. This efficiency is not just incremental but rather monumental when compared to the inference times of its contemporaries.

## 5 RELATED WORKS

### 5.1 Text-to-audio Generation

Text-to-Audio Generation is an emerging task that has witnessed notable advancements in recent years. For instance, Diffsound [49] leverages a pre-trained VQ-VAE [46] on mel-spectrograms to encode audio into discrete codes, subsequently utilized by a diffusion model for audio synthesis. AudioGen [20] frames text-to-audio generation as a conditional language modeling task, while Make-An-Audio [13], AudioLDM 2 [23], and TANGO [8] are all founded on the Latent Diffusion Model (LDM), which significantly enhances sample quality. However, a notable drawback of diffusion models lies in their iterative sampling process, leading to slow inference and restricting real-world applications. Unlike the diffusion model mentioned above, AudioLCM introduces a novel consistency-based models to support few-step and even one-step inference without sacrificing sample quality.

### 5.2 Consistency Models (CMs)

Consistency Models (CMs) [40] have emerged as a promising approach for efficient sampling without compromising quality. They leverage consistency mapping to enable fast one-step generation, as demonstrated in various tasks such as image generation [28], video generation [48], and audio synthesis [2, 51]. For example, Luo et al. [28] propose the latent consistency models demonstrates the potential of CMs to generate higher-resolution images. And Bai et al. [2] apply the consistency model into text-to-sound generation. However, their potential in high-quality text-to-audio generation has not been fully explored. In this study, we incorporate CMs into the text-to-sound and text-to-music generation tasks to uncover their capabilities.

## 6 CONCLUSION

Latent Diffusion Models (LDMs) have made significant strides in generative tasks but are hindered by slow generation due to their iterative sampling process. In this work, we introduced AudioLCM, a latent consistency model tailored for efficient and high-quality text-to-audio generation. Unlike prior approaches that addressed noise removal through iterative processes, AudioLCM integrated Consistency Models into the generation process, facilitating rapid inference through a mapping from any point at any time step to the trajectory's initial point. To overcome the convergence issue inherent in LDMs with reduced sample iterations, we proposed the Guided Latent Consistency Distillation with a multi-step ODE, which shortened the time schedule from thousands to dozens of steps while maintaining sample quality. Furthermore, we incorporated the methodologies of LLaMA and Diffusion transformer to enhance the performance and support variable-length generation. This architecture supported stable and efficient training, ensuring robust performance in text-to-audio synthesis. Experimental results on text-to-audio generation and text-to-speech synthesis tasks demonstrated that AudioLCM required only 2 iterations to synthesize high-fidelity audios, while maintaining sample quality competitive with state-of-the-art models using hundreds of steps. AudioLCM enabled a sampling speed of 333x faster than real-time on a single NVIDIA 4090Ti GPU, making generative models practically applicable to text-to-audio generation deployment. Our extensive preliminary analysis showed that each design in AudioLCM was effective.

## ACKNOWLEDGEMENTS

This work was supported by the National Key R&D Program of China under Grant No.2022ZD0162000.

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
