# OpenReview forum: "EchoAudio: Efficient and High-Quality Text-to-Audio Generation with Minimal Inference Steps"
_acmmm.org/ACMMM/2024/Conference — MM2024 Poster_

### Official Review · Reviewer_vDB2 · 2024-05-16

**Rating:** 3
**Confidence:** 3

**Summary:**

This paper proposes EchoAudio which is a new model designed for efficient and high-quality text-to-audio generation. The model addresses the computational challenges of diffusion models by integrating consistency models (CMs) into the generation process, which allows for rapid inference.
The paper also integrated some techniques like pre-layer norm, rotary embedding and SwiGLU from LLaMA into the foundational framework of transformers to optimize the performance of transformer-based neural network architectures.

Experimental results show that EchoAudio needs only 2 iterations to synthesize high-fidelity audios, maintaining sample quality competitive with teacher models using hundreds of steps. EchoAudio enables a sampling speed of 333x faster than real-time on a single NVIDIA 4090Ti GPU.

**Strengths:**

1. Apply the CM to audio generation tasks.
2. The generation inference speedup up to 333X seems impressive.

**Limitations:**

1. Insufficient novelty. CM was proposed for image generation. It was later also used in audio generation e.g. arXiv:2309.10740
2. The baselines and teachers looks weak on audiocaps evaluation data. Audiobox got FAD 0.78 much lower than the teacher model in the paper. Additionally audiobox uses 64 NFE.
3. The paper mentioned about the techniques from Llama. But there isn't ablation study to show the effectiveness these techniques to the audio generation tasks.

**Suitability:**

3

---

### Official Review · Reviewer_3eiD · 2024-05-24

**Rating:** 4
**Confidence:** 3

**Summary:**

The authors introduce EchoAudio, a model for efficient text-to-audio generation. Unlike previous approaches, EchoAudio integrates Consistency Models (CMs) into the generation process, allowing for rapid inference. To address convergence issues, the authors propose Guided Latent Consistency Distillation with a multi-step ODE solver. Additionally, they integrate advanced techniques from LLaMA into transformer-based neural network architectures to support stable and efficient training for text-to-audio synthesis.

**Strengths:**

EchoAudio introduces a unique approach by integrating Consistency Models (CMs) into the text-to-audio generation process. This novel integration allows for rapid inference. The presentation of EchoAudio is clear and well-structured. The authors systematically describe the problem, propose their solution, and validate their approach with detailed experiments.

**Limitations:**

While EchoAudio introduces the novel integration of Consistency Models (CMs) and a multi-step ODE solver, some model aspects build upon existing techniques without substantial innovation.

**Suitability:**

3

---

### Official Review · Reviewer_EtVK · 2024-05-24

**Rating:** 4
**Confidence:** 2

**Summary:**

The author proposes an audio generation model that integrates Consistency Models to decrease the number of NFE steps required for the diffusion model generation process. To tackle the convergence problem, the author further suggests a distillation method with a multi-step Ordinary Differential Equation (ODE) solver. The author also includes the mature architecture of the Llama model to enhance the model's structure. This results in a 2 NFE step student model that enables real-time generation for deployment.

**Strengths:**

1. The results are indeed impressive. The teacher model, with its improved architecture borrowed from Llama, already exhibits low FAD/KLD. With the distillation training for the Consistency Models, the student model manages to maintain a low FAD/KLD while only requiring 2 iterations during inference.

2. Audio generation modeling is a trending research area. A low RTF paves the way for the real-time deployment of these generation models.

3. The author conducts an ablation study to demonstrate the design choices of the ODE solver and Classifier-free Guidance, as well as the benefits of a Llama-like model architecture.

4. In addition to the FAD score, the author also provides audio generation samples, demonstrating that the model's generation is acceptable and closely matches the teacher model's generation.

**Limitations:**

1. The paper has limited novelty. The main concepts, such as Consistency Models, ODE solver, distillation of diffusion models, classifier-free guidance, and the Llama architecture, have already been explored in the image generation area or LLMs. The authors are primarily combining these elements in audio generation models.

2. For a fair comparison of quality and inference speed, the authors could include or discuss each model's size in the main table or abalation study. Is the teacher/student model superior simply because it's larger in scale? Moreover, training efficiency is another metric that the authors could compare, in addition to inference efficiency, especially considering the cost of training the teacher model and the distillation training.

3. Will the code/model be open-sourced if accepted?

**Suitability:**

3

---

### Official Review · Reviewer_EQaB · 2024-05-25

**Rating:** 3
**Confidence:** 3

**Summary:**

This paper presents EchoAudio, a novel consistency-based model designed for efficient and high-quality text-to-audio generation. By integrating Consistency Models (CMs) into the generation process and employing a multi-step Ordinary Differential Equation (ODE) solver, the authors aim to address the computational burden associated with Latent Diffusion Models (LDMs). The proposed method claims to achieve high-quality audio generation with significantly reduced inference steps. Experimental results demonstrate that EchoAudio achieves competitive performance with state-of-the-art models while being substantially faster.

**Strengths:**

1. Well-Written: The paper is clearly written and easy to follow, with a well-structured presentation of the proposed method and experimental results.

2. The method proposed by the authors is novel, particularly in how it integrates Consistency Models with a multi-step ODE solver to enhance the efficiency of text-to-audio generation.

3. The experimental results are comprehensive, showing that EchoAudio achieves competitive quality in text-to-audio generation with a much higher efficiency, which is a significant contribution to the field.

**Limitations:**

1. The related work section does not adequately cover existing works on consistency models in the audio and speech generation domains. Specifically, it lacks references to [1] and [2], which are closely related to the proposed method. The authors need to properly cite these works and discuss their relevance to the proposed method.

2. The paper states, "However, their potential in high-quality text-to-audio generation has not been fully explored." This claim is misleading as [2] also applies consistency models to text-to-audio generation. The authors should clarify the main differences and contributions of their method compared to this existing work and provide a thorough comparison in the experimental section.

3. Baseline Comparisons: The proposed method is claimed to be independent of the teacher model architecture. However, the authors have only conducted experiments using a single baseline, Make-An-Audio 2. It would be more convincing if the authors tested their method with multiple baselines to fully validate its effectiveness and generality.

[1] CoMoSpeech: One-Step Speech and Singing Voice Synthesis via Consistency Model, ACMMM 23
[2] ACCELERATING DIFFUSION-BASED TEXT-TO-AUDIO GENERATION WITH CONSISTENCY DISTILLATION

**Suitability:**

3

---

### Meta-Review · Area_Chair_c7gA · 2024-07-01

**Recommendation:** Accept (Poster)
**Confidence:** 4

**Metareview:**

Initially reviewers have mixed scores. Some reviewers have the concerns on the novelty of the proposed method, as well as the lack of some strong baseline comparisons. After the author rebuttal, most of the concerns are addressed. All reviewers changed their scores to borderline accept. The AC agrees with the reviewers, and thinks the work has enough contributions, thus recommending accept for this submission.